# Interplay between LncRNAs and microRNAs in Breast Cancer

**DOI:** 10.3390/ijms24098095

**Published:** 2023-04-30

**Authors:** Heidi Schwarzenbach, Peter B. Gahan

**Affiliations:** 1Department of Gynecology, University Medical Center Hamburg-Eppendorf, 20246 Hamburg, Germany; 2Fondazione “Enrico Puccinelli” Onlus, 06126 Perugia, Italy

**Keywords:** breast cancer, long non-coding RNAs, microRNAs, cancer detection rates

## Abstract

(1) Although long noncoding RNAs (lncRNAs) are known to be precursors of microRNAs (miRNAs), they frequently act as competing endogoneous RNAs (ceRNAs), yet still their interplay with miRNA is not well known. However, their interaction with miRNAs may result in the modulation of miRNA action. (2) To determine the contribution of these RNA molecules in tumor resistance to chemotherapeutic drugs, it is essential to consider not only the oncogenic and tumor suppressive function of miRNAs but also the impact of lncRNAs on miRNAs. Therefore, we performed an extensive search in different databases including PubMed. (3) The present study concerns the interplay between lncRNAs and miRNAs in the regulatory post-transcriptional network and their impact on drugs used in the treatment of breast cancer. (4) Consideration of this interplay may improve the search for new drugs to circumvent chemoresistance.

## 1. Introduction

Breast cancer (BC) is considered to be a widespread cancer in women aged 20–50 years and has been categorized into four phenotypically distinct subtypes: luminal A, luminal B, basal-like, and HER2 tumors. These are determined by either the presence (+) or absence (−) of specific molecular markers for either estrogen (ER) or progesterone receptors (PR) and human epidermal growth factor 2 (HER2): luminal A (ER+, PR high, HER2), luminal B (ER+, PR low, HER2−), HER+ (ER−, PR−, HER2+), and triple negative (ER−, PR−, HER2−). The quantification of the proliferation marker Ki-67 also contributes to this classification. The luminal A subtype has low Ki-67 levels, while the luminal B subtype has high Ki-67 levels, and HER2+ has either of these Ki-67 value. The ER status is an important predictive marker, and its loss leads to the resistance to endocrine treatment in BC patients. If BC cells are positive for all three receptors, cell proliferation is controlled by the ER and HER2. Basal-like tumors are mainly triple negative (TNBC) and account for 10–15% of all invasive BC types. The basal-like and HER2+ subtypes are more aggressive than luminal tumors. On the basis of the different clinical landscapes, these subtypes have different outcome responses to specific therapies [1].

A number of chemotherapeutic compounds have been developed to treat BC with a certain success. However, a downside to this approach concerns the development of resistance to treatment by such compounds and radiotherapy. Identifying the mechanism of cancer detection is made difficult due to the variety of possible processes involved in its initiation [2].

Thus, the tumor microenvironment itself can modulate drug flow causing a reduced drug uptake and multidrug resistance [3]. Membrane transport of anti-breast cancer drugs across the plasma membrane occurs mainly via two mechanisms. Anthracyclines, taxanes, platinum-based drugs, and oxazaphosphorines enter cells by passive diffusion across the plasma membrane whilst paclitaxel, docetaxel, cisplatin, and carboplatin can employ steroid hormone transporters. In addition, cisplatin and carboplatin have both been found to use copper transporters with cisplatin also passing via organic cation transporting polypeptides (Muley et al. [4]). Hence, any failure in these mechanisms of membrane transport result in loss of drug availability.

Solid tumor tissue hypoxia can result in a number of pathological changes linked to tumor growth. The reduced oxygen tension results in tumor biology modification and a number of hypoxia-inducible factors, leading to resistance induction involving a range of anti-cancer therapies [5,6,7]. Additionally, genetic modifications through either the methylation or demethylation of DNA can lead to cancer drug resistance (CDR) in both solid tumors [8] and experimental tumor cells in vitro [9,10].

RNA methylation can also have an effect leading to CDR. Thus, RNA N^6^-methyladenosine (m6A) modification occurs in ≈25% of mRNAs at the transcriptome-wide level. The proteins involved in either the upregulation or downregulation of the methylation process include RNA m6A methyltransferases, demethylases, and binding proteins. Among the various tumor aspects affected is the induction of CDR [11]. For example, the primary m6 A methyltransferase METTL3 is significantly downregulated in human sorafenib-resistant hepatocellular carcinoma. It was shown that in METTL3 depletion, under hypoxic conditions, sorafenib resistance occurred in cultured HCC cells [12].

More recently, noncoding RNAs have also been considered to be involved with important roles in CDR. They can generally be classified as micro-RNAs (miRNAs) and long non-coding RNAs (lncRNAs). miRNAs mediate gene expression regulatory effects at the post-transcriptional level and can occur either as freely circulating molecules or bound within endosomes [13]. Thus, overexpression of miR-130b leads to adriamycin resistance in BC cells by targeting PTEN [14]. lncRNAs are also present as either freely circulating molecules or bound within exosomes [13,15] and so correlate with CDR.

In this review, the interaction of miRNAs and lncRNAs are considered in light of CDR during BC treatment (Figure 1).

The information contained in mRNA is translated into protein by the ribosome. When miRNA binds to mRNA, the translation process is inhibited. However, the binding of lncRNA to miRNA can modulate this process and possibly abrogate it.

By binding of the lncRNA to the miRNA, the sequestered miRNA is no longer able to bind to the mRNA any more so that the translation of the mRNA performed by the ribosome is not disturbed. This interplay is involved in numerous processes, such as DNA methylation, apoptosis, DNA damage repair, epithelial–mesenchymal transition (EMT), and multidrug resistance (MDR). It is dependent on the concentration and character of the lncRNA.

## 2. Results

### 2.1. LncRNAs and miRNAs

Some 93% of the human genome DNA can be transcribed into RNAs, with <2% of such nucleotide sequences coding for proteins. The remaining 98% are classified as non-coding RNAs (ncRNAs) that partially or totally lack the necessary coding to produce proteins. Such ncRNAs are subdivided into a number of different forms on the basis of their length, localization, and/or function. Two such ncRNAs, miRNAs and lncRNAs, play a role in the induction of CDR through both their single action and their interplay [16] (Figure 1).

#### 2.1.1. How Are LncRNAs Produced?

LncRNAs comprise 200 or more nucleotides lacking open reading frames [16,17] and can be grouped according to their position relative to the protein-coding genes. As a result, they can be approximately divided into antisense, enhancer, bidirectional, intronic transcript lncRNAs, and large intergenic non-coding RNAs [18]. The biogenesis of lncRNA parallels that of mRNA since both processes involve RNA polymerase II. Furthermore, lncRNAs may regulate gene expression [19] and evolve either with or without polyadenylation [17], alternative cleavage, polyadenylation, as well as splicing [20], thus resulting in different isoforms from the same locus [17,19].

#### 2.1.2. lncRNAs Functions

LncRNAs can affect, e.g., housekeeping functions in addition to specialized functions, such as genomic imprinting and dosage compensation [17,21]. In addition, lncRNAs have been shown to regulate gene expression via multiple mechanisms—at the epigenetic, transcriptional, and post-transcriptional levels [22]. A particular feature of lncRNAs is their capacity to cross-talk with mRNA by competitively binding to shared miRNAs—a sponging effect [16,23]. LncRNAs have been shown to participate in the regulation of a variety of cell activities through interaction with other RNAs, DNAs, or proteins [22]. These include cell differentiation, proliferation, invasion, apoptosis, and autophagy.

#### 2.1.3. How Are miRNAs Are Produced?

As with mRNA, all miRNA host genes form primary RNA transcripts by RNA polymerase II transcription [24]. Subsequently, 21–23 nucleotide miRNA molecules result from typical splicing, capping, and polyadenylating, as occurs with protein-coding mRNAs [25]. Maturation of the stem-looped, precursor miRNA is achieved by cleavage through the action of the enzyme Drosha followed by further cleavage through the action of the enzyme Dicer. Following a number of maturation steps, the mature miRNA may then be integrated into the RISC complex, an Argonaut containing an RNA–protein complex, for its role in mRNA breakdown [26]. Some miRNAs may be produced by alternative routes in which the Dicer step is omitted. A comparison of total miRNA with RISC protein copy number implies that a major fraction of the miRNAs is free in the cell—both in the nucleus and cytoplasm. Of these, a number will be secreted, often in exosomes, for transfer to other recipient cells where they will participate in various cellular events [27].

#### 2.1.4. miRNA Functions

Various studies have been performed on the interactions between lncRNA and miRNA and their relationship with mRNA in addition to those already mentioned above. Their interactions can result in their competition for the same mRNA and miRNA-initiated RNA breakdown, as well as miRNA derivation from lncRNAs plus and a role for lncRNAs being decoys for miRNAs [28]. The decoy effect by lncRNAs is likely to be the major effect in cancer through their acting like sponges in binding miRNAs [23], preventing the latter from completing their regulatory function, i.e., they act as positive regulators of mRNA transcripts. Much of the capturing of miRNAs by lncRNAs occurs primarily towards the 3′ ends of the miRNAs, i.e., similar to the Argonaut binding sites, and they are termed miRNA response elements [29].

#### 2.1.5. Mechanism of LncRNA–miRNA Interactions

In 2010, Wang et al. [30] first identified the lncRNA-associated endogenous RNA (ceRNA) mechanism in liver cancer, where lncRNA HULC served as a sponge for miR-372, inhibited its activities, and consequently reduced the repression of PRKACB (pigmented spots, myxomas, pituitary adenomas), which in turn induced phosphorylation of CREB. A year later, the ceRNA network hypothesis was suggested by Salmena et al. [23], which implied that lncRNAs can act as ceRNAs, also called miRNA sponges. The study by Hansen et al. [31] in the same year confirmed the ceRNA hypothesis and described the interaction between circRNA and miRNA. In this respect, these scientists identified circRNA ciRS-7, which originated from cerebellar-degeneration-related protein 1 antisense transcript (CDR1AS) as a sponge for miR-7 (ciRS-7) via the miRNA-dependent binding to argonaute (AGO) proteins.

LncRNAs can interact with miRNAs via miRNA response elements (MREs), specific sequences within secondary structures. This interplay may lead to a decrease in the miRNA levels, an impairment of miRNA activity, and an interference of the gene regulation. However, the efficiency of the ceRNA function is also based on the relative levels of lncRNAs and miRNAs. Hence, changes at the ceRNA level are critical since they result in either the potentiation or the attenuation of the functions of miRNAs on their mRNA target genes, as well as changes in the signaling pathways. Furthermore, due to their interaction with the aberrant levels of lncRNAs in cancer cells, miRNAs may be more susceptible to degradation than lncRNAs [32]. While lncRNAs can also be precursors of miRNAs and so can regulate miRNA biogenesis at different steps, miRNAs can regulate the stability and half-life of lncRNA [33]. In addition, lncRNAs can compete with miRNAs for binding to the target sites of mRNA, as demonstrated by antisense transcript for the β-site amyloid precursor protein cleaving enzyme 1. This antisense transcript binds to its sense partner in the miR-485-5p recognition site, thus abrogating its function in Alzheimer’s disease [34]. As described for lncRNA H19, some lncRNAs can exert multiple functions. H19 acts as a molecular sponge for let-7 and is also a precursor of miR-675, mediating muscular differentiation and regeneration [35].

The ceRNA activity results in the formation of a large-scale RNA regulatory network across the transcriptome. This implies that such a network can result in the formation of a new cellular language. Thus, a much-expanded network of functional genetic information has been developed and is now being shown to play an important role in pathological conditions. This approach has been adopted a range of studies on cancer in general, such as hepatocellular carcinoma [36] and a range of BC detection [37,38,39], as discussed below.

### 2.2. NcRNA Delivery Systems

There are a number of modes of ncRNA transport available for bringing ncRNAs to the tumor site [40].

#### 2.2.1. Exosomes

Exosomes are the first line of attack in that they are naturally occurring structures that, on entering cells, are able to avoid endosomal uptake. This is likely due to the presence of the exosomal membrane protein CD47 that initiates the signal leading to inhibition of phagocytosis. In vivo, they are naturally targeted nucleic acid transporting structures, allowing the movement of both DNA and RNAs from healthy to healthy cells, healthy to tumor cells, and tumor to healthy cells [41]. Thus, preparation and use of exosomes carrying the specific tumor cell marker should enable delivery of the lncRNA to the specific tumor. As in the case of cancer cell-derived exosomes, mesenchymal stem cell (MSC)-derived exosomes have also been shown to have specific tumor targeting capabilities [42,43]. This may be due to the intimate interaction between MSCs and tumor tissues.

Large-scale production of exosome-mimicking nanovesicles can occur as the result of serial extrusion from cells [44,45], resulting in a vesicle yield 150-fold higher than that of exosomes. Such vesicles have similar physicochemical properties and anti-cancer effects to exosomes [45]. The relevant RNA can be introduced into the vesicles by, e.g., electroporation. However, incubation appears to be the simplest way to load lncRNA into exosomes. Without membrane modification, RNAs have been shown to diffuse into exosomes along a concentration gradient [46]. To overcome the inefficient loading by this method, Munagala et al. suggested the addition of cationic polyethylenimine that greatly improves the RNA transfection efficiency when compared with both electroporation and direct exosomal transfection. An alternative approach to the simple diffusion method involves modification of RNAs with cholesterol to form hydrophobic RNAs. These can then insert into the exosomal lipid bilayer [47,48]. Such an approach improves both the loading efficiency and stability of the RNA [49], without influencing either the physical or functional exosomal properties.

#### 2.2.2. Cell Membranes

Cell membranes offer an alternative approach since, like exosomes, they have an intrinsic targeting system as well as the ability to avoid the immune system. Two major sources of such membranes are cultured tumor cells [50] and erythrocytes [50,51]. Such membrane fractions are extruded through polycarbonate membranes to yield cell membrane vesicles. These can then be combined with nuclear proteins. Unfortunately, such membrane fractions need modification to if they are to avoid endosomal uptake during RNA delivery [50].

#### 2.2.3. Liposomes

Liposomes are the earliest structures to be employed to transport molecules into cells. They are comprised of phospholipids, e.g., phosphatidylcholine, phosphatidylethanolamine, phosphatidylserine, and phosphatidylglycerols, being stabilized by the addition of cholesterol. However, they have not been very efficient in the transfer of nucleic acids into cells and have been replaced by lipid nanoparticles.

#### 2.2.4. Lipid Nanoparticles

The FDA has approved lipid-based structures that may be exploited for the delivery of RNAs to tumor cells. These contain four basic components, namely, cholesterol, a helper lipid, a PEG lipid, and a cationic or ionizable lipid. Such polymeric nanoparticles employ acationic amine groups that attach the anionic phosphodiester backbone of the particular RNA. Alternatively, dendrimers may be exploited to transfer the RNA. These are polymeric structures with a specific number of molecules arising from the center [52].

### 2.3. Effective Drugs and Important Signaling Pathways in Breast Cancer

Adjuvant or neoadjuvant therapies with the drugs, as described in Table 1, prevents disease recurrence in many BC patients. However, a significant number of patients acquire CDR. Since ER and HER2 are important molecules in the survival and proliferation of BC cells, research has focused on exploring the HER2- and ER-activated signaling pathways. An important aspect for CDR is the complexity of the signaling network and the crosstalk between signaling pathways as well as HER2 and hormone receptors. Relevant signaling pathways in BC are phosphoinositide-3- kinase/Akt/mammalian target of rapamycin (PI3K/AKT/mTOR) and mitogen-associated protein kinase (MAPK)/RAS/RAF pathways stimulated by hormone receptors and HER2, which are also affected by the interplay of lncRNAs and miRNAs. CDR of BC is mainly responsible for the poor prognosis and short survival rate of BC patients. Although CDR complicates the treatment strategies of BC, some chemotherapeutic, endocrine, and targeted drugs, such as anthracyclines, taxanes, cisplatin, and 5-fluorouracil (5-FU), have significantly improved the quality of life and overall survival of BC patients [53] (Table 1).

#### 2.3.1. Anthracyclines

Anthracyclines including the most used doxorubicin, daunorubicin, and epirubicin are a group of antibiotics. These agents are very effective in the treatment of BC at all tumor stages, but they also cause cardiotoxicity that limits its clinical application. The induced failure of the heart to efficiently pump blood through the body is doxorubicin dose dependent. The age of the patients also plays a role in vulnerability since years after completed treatment younger patients still have a higher risk than older ones for developing cardiotoxicity and heart failure. The anticancer properties of doxorubicin are based on its structure. It is composed of a tetracyclic ring, two hydroxyl groups and a sugar attached to the ring with a glycosidic bond that causes its hydrophilic properties. Doxorubicin forms complexes with DNA by intercalating between the DNA base-pairs, resulting in bidirectional transmission of positive torsion. This inhibits topoisomerase IIα activity that unlinks intertwined DNA strands. Blocking of topoisomerase IIα by doxorubicin stabilizes the DNA-topoisomerase IIα complex leading to DNA double-strand breaks and cancer cell death [54].

#### 2.3.2. Tamoxifen

Endocrine therapy includes the use of a selective ER modifier or an aromatase inhibitor. Tamoxifen is approved for pre-menopausal women. It is an anti-estrogen that blocks the estrogen receptor and is, in particular, commonly applied for the ER+ BC subtype, accounting for more than 70% of all BCs. In the majority of BC patients, it is devoid of major side effects. However, metastatic ER+ patients respond poorly to tamoxifen therapy and frequently display an increased dose- and time-developed resistance to it [55].

#### 2.3.3. Tyrosine Kinase Inhibitors

Due to a gene amplification, HER2 tumors overexpress the receptor tyrosine kinase HER2 and are effectively targeted by using first-line humanized, monoclonal antibodies, including trastuzumab and pertuzumab or, subsequently, second-line tyrosine kinase inhibitors, such as lapatinib or neratinib. Due to a loss of the hormone receptors and HER2, TNBCs are treated with cytotoxic chemotherapy. Trastuzumab is a HER2-targeting humanized monoclonal antibody that blocks the activation of HER2 leading to lysis of the cancer cells. It is selective for the treatment of HER2 BC patients, but also for the metastatic stomach cancer. There is a high correlation of HER2 overexpression with BC metastasis as well as poor prognosis. Among others, it may cause cardiotoxicity [56].

#### 2.3.4. Taxanes

Taxanes, including the most commonly used paclitaxel and docetaxel, are a class of antineoplastic agents which are involved in angiogenesis, apoptosis, cell motility, invasiveness, and metalloproteinase production; they are applied for the treatment of a wide variety of cancers. Along with anthracyclines, they are the standard therapy for TNBC patients. These cytotoxic agents inhibit the depolymerization of tubulin microtubules consisting of α and the β tubulin dimers, the main components of the cytoskeleton. Paclitaxel selectively binds to the β subunit and alters its assembly so affecting the formation of a mitotic spindle. This leads to the arrest of the G2/M phase of the cell cycle with inhibition of mitosis followed by cell death. Immediate hypersensitivity reactions occur in 5–10% of patients. Excessive immune reaction of the body to allergenic substances is a well-known problem in the treatment of cancer patients, but it is not an exclusion factor for the therapy. Most chemotherapy drugs can cause allergic reactions. However, almost all patients that get these reactions can be re-exposed to taxanes either through desensitization or challenge [57].

#### 2.3.5. 5-Fluorouracil

5-Fluorouracil (5-FU) is a classic chemotherapeutic drug, used extensively for different cancers, e.g., colorectal cancer. It is a heterocyclic organic compound with a pyrimidine backbone and two carbonyl groups at the 2- and 4-positions and a fluorine at the 5-position. 5-Fluoro-2′-deoxyuridine 5′-monophosphate (FdUMP) is an active metabolite of 5-FU and inhibits thymidylate synthase, preventing incorporation of 2′-deoxythymidine 5′-monophosphate (dTMP) into DNA via 2′-deoxythymidine 5′-triphosphate (dTTP). This results in the accumulation of dUMP (a metabolite of uracil) which is then incorporated into DNA, causing DNA damage. A frequent side effect is the increased risk of becoming infected due to a drop in blood leukocytes levels [58].

#### 2.3.6. PI3K/Akt/mTOR

As reviewed in detail by Paplomata and O’Regan [59], PI3K/Akt/mTOR is a major signaling pathway responding to nutrients, hormones, and growth factor stimulation. Briefly, the PI3Ks phosphorylate phosphatidylinositols, which in turn phosphorylate the serine/threonine kinase Akt, influencing the cancer cell cycle, survival, and growth. Phosphatase and tensin homolog deleted from chromosome ten (PTEN), an important tumor suppressor, has an opposite and dephosphorylating effect on phosphatidylinositols. The serine/threonine protein kinase mTOR is located downstream of PI3K and Akt. In BC, the PI3K/AKT/mTOR pathway leads to cell growth and tumor proliferation and is involved in endocrine resistance. The losses of PTEN and PIK3CA mutations are among the most common aberrations in BC, and it is assumed that tumors with PIK3CA mutations are more sensitive to inhibitors of the PI3K pathway. The pathway has been correlated with resistance to endocrine therapy, HER2-directed therapy, and cytotoxic therapy. A combination of agents targeting multiple steps of the pathway is recommended since activation of pathways, such as KRAS and MEK, can escape the treatment with a specific agent, leading to resistance. Furthermore, Akt signaling plays a cellular defense role against chemotherapy, enhancing multi-drug resistance. Overactivation of Akt and its upstream and downstream regulators in resistant BC cells is considered to be a major potential target for new anti-cancer therapies in BC.

Both miRNAs and lncRNAs are involved in this pathway. For example, miR-182 reversed the trastuzumab resistance of BC cells via targeting c-MET and its downstream PI3K/AKT/mTOR pathway [60]. In contrast, miR-221 is upregulated in trastuzumab-resistant BC cells and directly represses PTEN to cause increased motility and invasiveness of HER2+ BC cells. Suppression of miR-221 or restoration of PTEN reversed the malignant phenotypes of HER2+ BC [61]. In addition, silencing of lncRNA HOTAIR significantly decreased the phosphorylation of PI3K, AKT, and mTOR and attenuated the resistance of BC cells to doxorubicin by inhibiting the PI3K/AKT/mTOR pathway [62]. LncRNA GAS5 also regulates the PI3K/AKT/mTOR pathway as well as Wnt/β-catenin and NF-κB signaling and can increase the sensitivity to multiple drugs [63].

#### 2.3.7. MAPK/RAS/RAF

As described in a review by Cargnello and Roux [64], the PI3K/AKT/mTOR and MAPK/RAS/RAF pathways are known to interact with each other at several node points, suggesting that dual blockade of both pathways may be required for a successful anticancer therapy. The MAPK/RAS/RAF pathway plays a critical role in, e.g., proliferation, differentiation, development, transformation, and apoptosis. The cascade is activated by growth factors and cytokines binding to receptor tyrosine kinases, G-protein-coupled receptors, and non-nuclear activated steroid hormone receptors, involving a series of protein kinase cascades (e.g., MAPKs). The most common BC drugs that regulate or inhibit MAPK/RAS/RAF pathway include the farnesyltransferase inhibitors, sorafenib, and vemurafenib.

Dysregulation of this pathway caused by ncRNAs have been reported along with its implications in cancer treatment failure. For example, lncRNA AWPPH and miR-21 overexpression led to cell proliferation during carboplatin treatment of TNBC [65]. In addition, miR-1275 plays a crucial role in regulating this signaling pathway and may predict the prognosis of BC patients [66].

#### 2.3.8. WNT/β-Catenin

Harold Varmus and Roel Nusse discovered and investigated the Wnt signaling pathway forty years ago [67,68]. The Wnt signaling pathway comprises both a canonical and non-canonical pathway. The former signaling is initiated when WNT ligands bind to the transmembrane receptor Frizzled (FZD) protein, whereby the single pass low-density lipoprotein-receptor-related protein (LRP) and 5/6 membrane proteins act as co-receptors. This leads to expression of the transcriptional co-activator β-catenin that regulates key developmental gene expression programs. Signaling by the Wnt family of secreted glycolipoproteins directs cell proliferation, cell polarity, and cell fate determination during embryonic development and tissue homeostasis. In addition, the Wnt/β-catenin pathway plays a crucial role in the development of the mammary gland during embryogenesis and during pregnancy. It is one of the most deregulated pathways in BC in which it is associated with initiation, progression, metastasis, and maintenance of cancer stem cells and CDR. Therapeutics that target the WNT signaling pathway are combinatorial therapies with either tyrosine kinase inhibitors or immune checkpoint blockers. Resistance to these tyrosine kinase inhibitors is caused by multiple mechanisms, such as acquired drug-resistant mutations in targeted tyrosine kinases, epithelial–mesenchymal transition (EMT), and activation of other tyrosine kinase signaling cascades so as to bypass targeted tyrosine kinases [69]. In addition, the Wnt/β-catenin axis, through the Snail protein, promotes the expression of miR-125b and chemoresistance in BC stem cells [70].

#### 2.3.9. TGF-β/Smad

TGF receptor activation induces the phosphorylation of SMADs that then form a complex with a co-mediator SMAD. In co-operation with different transcription factors and co-factors, these complexes control the transcription of hundreds of genes. Thus, the TGF-β pathway can coordinate various cellular processes, including cell growth, differentiation, cell migration, invasion, and extracellular matrix remodeling. In BC, TGF-β has opposing roles by acting as both a tumor suppressor in the initial tumor stage but stimulating invasion and metastasis at later stages. The latter events are through changes in the expression of cell–cell adhesion molecules and secretion of metalloproteinase. In normal cells and early stage BC, TGF-β signaling inhibits cell growth, but as cells acquire mutations in TGF-β and its downstream effectors, TGF-β signaling promotes tumor growth. Moreover, TGF-β activation can lead to multiple cellular responses mediating drug resistance to antiestrogens, e.g., tamoxifen, the most commonly prescribed antiestrogen [71,72].

LncRNA LINC00665 participates in the TGF-β signaling pathway besides five other signaling pathways including the Wnt/β-catenin, NF-κB, PI3K/AKT, and MAPK signaling pathways. Aberrant expression of LINC00665 in BC is associated with poor prognosis and affects cisplatin-paclitaxel treatment [73].

### 2.4. Involvement of lncRNAs and miRNAs in Signaling Pathways and Drug Resistance

Receptors, protein tyrosine kinases, phosphatases, proteases, signaling pathways, miRNAs, and lncRNAs are potential therapeutic targets in BC treatment.

LncRNAs and miRNAs regulate multiple signaling pathways in BC, including the pathways described above and as shown in Table 2 below. Depending on the binding activity of these ncRNAs to the signaling molecules, their involvement leads to either activation or repression of the pathways. However, long-term medication modulates these pathways generating CDR. For example, some lncRNAs that are regulated by WNT and/or participate in WNT pathway modulation and outcome play a role in cancer. H19 is such a molecule that modulates the WNT/β-catenin signaling pathway in several ways to possibly promote CDR. It acts as a molecular sponge, inhibiting let7-mediated downregulation of c-myc expression [74]. Moreover, it activates the WNT/β-catenin pathway, promoting osteoblast differentiation by functioning as a lncRNA–miRNA–mRNA ceRNA network for miR-141 and miR-22, which are negative regulators of osteogenesis [75]. Furthermore, increased MALAT1 expression is associated with poor recurrent-free survival in tamoxifen-treated ER+ BC patients, predicting the answer to the endocrine treatment [76].

### 2.5. Involvement of lncRNAs and miRNAs in Angiogenesis

Angiogenesis is the formation of new blood vessels from pre-existing ones, playing a role in physiological and pathological processes. Abnormal vessel growth is a hallmark of cancer, and VEGF is a central factor in this process [77]. In 1971, Sherwood et al. were the first scientists to identify tumor angiogenesis as a putative therapeutic target [78], leading to the development of anti-angiogentic therapies. Although numerous miRNAs and lncRNAs have been described in angiogenesis of BC, as far as we know, only a few articles deal with the interplay of miRNAs and lncRNAs in angiogenesis of BC [79].

In their study [80], Zhang et al. described lncRNA NR2F1-AS1 to promote BC angiogenesis in vitro and in vivo through the activation of the IGF-1/IGF-1R/ERK pathway. NR2F1-AS1 correlated with the vessel target CD31 and CD41 and was able to enhance the tube formation ability of HUVEC cells. In a zebrafish model, lncRNA NR2F1-AS1 facilitated the neovascularization and promoted the metastasis of BC cells, probably as a result of the lncRNA NR2F1-AS1 increased tumor vasculature. In addition, in a mouse model, lncRNA NR2F1-AS1 induced more tumor vessels and higher micro-vessel density in the tumor mass. Mechanistically, lncRNA NR2F1-AS1 increased IGF-1 expression through sponging miR-338-3p in BC cells followed by activation of the receptor of IGF-1 and ERK pathways. In an earlier study [81], these scientists found that lncRNA NR2F1 downregulated the expression of miR-200s, which in turn upregulated the expression of IL-8 in BC cells so inducing BC angiogenesis through the IL-8/lncRNA NR2F1/miR-200s/IL-8 loop.

In BC cells, Sun et al. [82] identified lncRNA LINC00968 as a ceRNA of miR-423-5p. Overexpression of LINC00968 inhibited BC cell proliferation, migration, and tube formation in vitro as well as tumor growth in vivo through inhibition of miR-423-5p, which downregulated PROX1. In contrast, LINC00968 increased PROX1 expression in vivo. Thus, LINC00968 may reduce angiogenesis in BC by upregulating PROX1 and reducing miR-423-5p.

Huang et al. [83] showed that MALAT1 promoted angiogenesis in BC. In MCF-7 cells, MALAT1 knockdown was able to significantly inhibit proliferation, migration, and tube formation, as well as increase miR-145 levels. In BC tissue, the expression of MALAT1 was inversely associated with miR-145. In addition, knockdown of MALAT1 reduced the expression of VEGF, promoting angiogenesis in BC by the interaction with miR-145.

In metastatic BC, NFAT5 promoted EMT and invasion of cells. Li et al. [84] revealed that this lncRNA induced the expression of the calcium-binding protein S100A4, facilitating the angiogenesis of breast epithelial cells by transcriptionally activating VEGF-C. NFAT5 was directly targeted by miR-568, which was in turn inhibited by the lncRNA HOTAIR. This epigenetic silencing led to methylation of histone H3K27 and demethylation of H3K4 on the miR-568 loci.

### 2.6. CeRNA Network in Breast Cancer

Given the development of the ceRNA concept (16), lncRNAs should be systematically analyzed as either potential tumor suppressors or oncogenes through their ceRNA function. In the ceRNA network, the relative concentration of ceRNAs and their miRNAs is essential. Changes in the ceRNA levels may either endorse or abrogate the mRNA repression by miRNAs. In physiological and pathological conditions, this on–and–off switch may be regulated in different developmental stages. Moreover, the effectiveness of a ceRNA to sequester miRNAs may depend on the number of miRNAs it is able to sponge, as well as the accessibility to miRNAs, which is influenced by its subcellular localization. Thus, this lncRNA–miRNA interaction may depend on the character, concentration, and subcellular distribution of these molecules that can be influenced by pathological processes. Our future task will be to understand this complex regulatory network, which consequences it has in cancer and when CDR is perturbed.

Numerous studies on the interplay between lncRNAs and miRNAs have been made, a few recent examples of which are described in the following paragraphs. Additional examples are also listed in Table 2 and are depicted in Figure 2.

**Table 2 ijms-24-08095-t002:** Interaction between lncRNAs and miRNAs in chemoresistant breast cancer.

lncRNA	miRNA	Drug	Molecule/Pathway	Ref.
RP11-70C1.3	miR-6736-3p	Anthracycline,	NRP1	Zhang et al. [85]
		taxanes		
CDR1-AS	miR-7	Cisplatin	REGγ	Yang et al. [86]
DLX6-AS1	miR-199b-5p	Cisplatin	Paxillin	Du et al. [87]
TMPO-AS1	miR-1179	Docetaxel	TRIM37	Ning et al. [88]
PTENP1	miR-20a	Doxorubicin	PTEN, PI3K/Akt	Gao et al. [89]
CBR3-AS1	miR-25-3p	Doxorubicin	MEK4/JNK1	Zhang et al. [90]
DNAJC3-AS1	miR-144	Doxorubicin		Ren et al. [91]
LINC00518	miR-199a	Doxorubicin	MRP1	Chang et al. [92]
		Vincristine		
		Paclitaxel		
XIST	miR-200c-3p	Doxorubicin	Anilin	Zhang et al. [93]
GAS5	miR-221-3p	Doxorubicin	Dickkopf 2,	Chen et al. [94]
			Wnt/β-catenin	
DLGAP1	miR-299-3p	Doxorubicin	WTAP	Huang et al. [95]
MALAT1	miR-570-3p	Doxorubicin		Yue et al. [96]
HSAT101069	miR-129-5p	Epirubicin	Twist1	Yao et al. [97]
AFAP1-AS1	miR-195/miR-545	Formononetin	CDK4, Raf-1	Wu et al. [98]
NEAT1	miR-211	5-FU	HMGA2	Li et al. [99]
PRLB	miR-4766-5p	5-FU	Sirtuin 1	Liang et al. [100]
CASC2	miR-18a-5p	Paclitaxel	CDK19	Zheng et al. [101]
AC073284.4	miR-18b-5p	Paclitaxel	DOCK4	Wang et al. [102]
NEAT1	miR-23a-3p	Paclitaxel	FOXA1	Zhu et al. [103]
LINC00511	miR-29c	Paclitaxel	CDK6	Zhang et al. [104]
FTH1P3	miR-206	Paclitaxel	ABCB1	Wang et al. [105]
H19	miR-340-3p	Paclitaxel	YWHAZ,	Yan et al. [106]
			Wnt/β-catenin	
GAS5	miR-378	Paclitaxel	SUFU	Zheng et al. [107]
MALAT1	miR-485-3p	Paclitaxel	P-GP, BCL2, BAX	Aini et al. [108]
DDX11-AS1	miR-497	Paclitaxel		Liang et al. [109]
UCA1	miR-613	Paclitaxel	CDK12	Liu et al. [110]
SNHG7	miR-3127-5p	Paclitaxel		Yu et al. [111]
TPT1-AS1	miR-3156-5p	Paclitaxel	Caspase 2	Huang et al. [112]
MEG3	miR-4513	Paclitaxel	PBLD	Zhu et al. [113]
NUDT3-AS4	miR-99s	Rapamycin	KT1/mTOR	Hao et al. [114]
UCA1	miR-18a	Tamoxifen	HIF1α	Li et al. [115]
DANCR, GAS5,	miR-29b-1/a	Tamoxifen		Muluhngwi et al. [116]
DSCAM-AS1,				
SNHG5, CRND				
CYTOR	miR-125a-5p	Tamoxifen	MAPK, SRF	Liu et al. [117]
ADAMTS9-AS2	miR-130a-5p	Tamoxifen	PTEN	Shi et al. [118]
DSCAM-AS1	miR-137	Tamoxifen	EPS8	Ma et al. [119]
ROR	miR-205	Tamoxifen	ZEB1, ZEB2	Zhang et al. [120]
MAFG-AS1	miR-339-5p	Tamoxifen	CDK2	Feng et al. [121]
HNF1A-AS1	miR-363	Tamoxifen	SERTAD3,	Li et al. [122]
		TGF-β/Smad		
FOXD3-AS1	miR-363	Tamoxifen	Trefoil factor 1	Ren et al. [123]
AGAP2-AS1	miR-15a-5p	Trastuzumab	CPT1	Han et al. [124]
UCA1	miR-18a	Trastuzumab	YAP1	Zhu et al. [125]
GAS5	miR-21	Trastuzumab	mTOR	Li et al. [126]
TINCR	miR-125b	Trastuzumab	Snail-1	Dong et al. [127]
OIP5-AS1	miR-381-3p	Trastuzumab	HMGB3	Yu et al. [128]

BCL2, B-cell lymphoma 2; BAX, Bcl2-associated X protein; CASC2, cancer susceptibility candidate 2; CDK2, cyclin-dependent kinases; CPT1, carnitine palmitoyl transferase 1; CYTOR, cytoskeleton regulator; DOCK4, dedicator of cytokinesis protein 4; EIF5A2, eukaryotic translation initiation factor 5A-2; EPS8, epidermal growth factor receptor pathway substrate 8; FTH1P3, ferritin heavy chain 1 pseudogene 3; 5-FU, 5-fluorouracil; HMGA2/B3, high-mobility group protein A2/B3; MALAT1, metastasis-associated lung adenocarcinoma transcript 1; MAPK, mitogen-associated protein kinase; NRP1, neuropilin 1; PBLD, phenazine biosynthesis-like domain-containing protein; P-GP, P-glycoprotein; PI3K, phosphoinositide 3-kinase; PTEN, phosphate and tensin analog; PRLB, progression-associated lncRNA in breast cancer; SRF, serum response factor; SUFU, suppressor of fused; TRIM37, tripartite motif-containing protein 37; mTOR, mammalian target of rapamycin; TUG1, taurine-upregulated gene 1; UCA1, urothelial cancer associated 1; WTAP, N^6^-methyladenosine (m6A) methyltransferase WT1-associated protein; YAP1, Yes-associated protein 1; YWHAZ; tyrosine 3-monooxygenase/tryptophan 5-monooxygenase activation protein zeta.

A detailed description of the interplay and drugs is given in Table 1 and Table 2 together with the references and explanations of the abbreviations.

#### 2.6.1. LncRNA H19 and Its Interaction with Numerous miRNAs

H19, the first discovered lncRNA, has key regulatory functions in tumor development and progression. This paternally imprinted gene is located close to the telomeric region of chromosome 11p15.5 that often harbors genetic alterations in tumors. It is ectopically expressed in human malignant tumors, where it functions as an oncogene, but it also exhibits anti-oncogenic features in some tumors. H19 as ceRNA binds to different miRNAs in numerous tumors and so influences the downstream activity of the sequestered miRNAs. In this respect, H19 may control cell proliferation, apoptosis, EMT, tumor progression, metastasis, and drug resistance, suggesting its important role in the lncRNA–miRNA–mRNA network in cancer [129].

As demonstrated by Wei et al. [130], H19 may also regulate gene expression, namely, of HER2 in gastric cancer. They found that H19 may function as a ceRNA to regulate HER2 expression by sequestering miRNA let-7c, which as a tumor suppressor negatively correlates with the expression of HER2. It might be interesting to know whether or not this interplay also occurs in BC due to the central importance of HER2-based diagnostic and therapeutic strategies in BC. In this respect, we also detected that the plasma levels of H19 are associated with early HER2+ BC [131].

The involvement of H19 with tumorigenesis and invasion is partly owed to its regulation of carcinogenic miR-675 through its precursor RNA being located in the first exon of the H19 gene [35,132]. In our study, we found that H19 and miR-675 concentrations were significantly higher and lower, respectively, suggesting a relationship between H19 and miR-675 in BC. H19 stimulated cell proliferation, inhibited apoptosis, and affected co-transfected miR-675 [131]. As reported by Vennin et al. [133], the aggressive phenotype of BC cells may depend on a higher cell proliferation and migration in vitro, as well as an increased tumor growth and metastasis in vivo by the interplay of H19 and miR-675.

In the study by Yan et al. [106], the levels of H19 were demonstrated to be increased in the BC paclitaxel-resistant MCF-7 cell subline. Bioinformatics tools and dual-luciferase reporter assays indicated that miR-340-3p was a potential target gene of H19 that acted as a ceRNA to promote BC cell proliferation, metastasis, and EMT by regulating tyrosine 3-monooxygenase/tryptophan 5-monooxygenase activation protein zeta (YWHAZ) and potentiating the Wnt/β-catenin signaling in BC cells. It seems that H19 also plays a role in the Wnt/β-catenin pathway in other tumors. The study by Ren et al. [134] showed that H19 can activate this signaling pathway by functioning as a sponge for miR-141 in colorectal cancer.

These data indicate that lncRNAs do not only interact with miRNAs but that they can also be precursors of miRNAs. Thus, some lncRNAs may have an opposite function: the formation of a miRNA and the subsequent inhibition of this miRNA [135].

#### 2.6.2. LncRNA HOTAIR and Its Interaction with Different miRNAs

In the same year, 2007, as the detection of the imprinted lncRNA H19 being a primary miRNA precursor of miR-675, HOTAIR was identified by Rinn et al. [136]. They revealed that this lncRNA forms the limit between two diametrical chromatin domains in the homeobox C cluster (HOXC) locus. It is transcribed in an antisense manner and is involved in chromatin dynamics as well as regulating gene silencing associated with histone methylation. In particular, the aberrant expression of HOTAIR plays an important role in BC by contributing to tumor progression. Moreover, it is a prognostic factor of BC metastasis. HOTAIR is a key regulator of chromatin status and a mediator of transcriptional silencing. In in vitro and in vivo BC models, estradiol agonists, bisphenol-A, and diethylstilbestrol are able to stimulate HOTAIR expression [137].

Cancer stem cells are responsible for maintaining tumor cell populations and play an essential role in cancer metastasis. HOTAIR was shown to inhibit miR-7 in BC stem cells, which in turn inhibited cell invasion and metastasis of BC stem cell xenografts, EMT, and BC stem cell density [138]. Furthermore, miR-34a seems to be an important target of HOTAIR regulation. In the study by Deng et al. [139], HOTAIR was the only overexpressed lncRNA-regulated proliferation, colony formation, migration, and self-renewal of cancer stem cells obtained from the BC MCF-7 cell line. The interaction with miR-34a contributed to these effects. Moreover, transcriptional control of miR-34a by HOTAIR influenced SOX2 (SRY (sex determining region Y)-Box 2), an essential stemness factor regulating the self-renewal capacity of cancer stem cells. Upregulation of HOTAIR also affected proliferation and colony formation in cancer stem cells by inducing p53 expression in the p53/p21 pathway.

In addition, there is an inverse correlation of HOTAIR with miR-141 expression in renal carcinoma cells. Whereas HOTAIR promotes proliferation and invasion of these cells, miR-141 suppresses these malignant effects. The binding of miR-141 to HOTAIR suppresses HOTAIR expression through a direct cleavage by Argonaute 2, RISC (RNA interference silencing complex) catalytic component 2 (AGO2), that usually cleaves target transcripts of miRNAs. Since, in contrast to HOTAIR, miR-141 is a tumor suppressor, its interaction with HOTAIR may have an impact on the process of malignant transformation, which may also include breast carcinogenesis [140].

Li et al. [62] showed lncRNA HOTAIR silencing to decrease cell proliferation as well as increasing apoptosis in both MCF-7 cells and doxorubicin-resistant MCF-7 cells. It also decreases phosphorylation of PI3K, AKT, and mTOR, indicating that the knockdown of this lncRNA-HOTAIR attenuated the resistance of BC cells to doxorubicin through inhibition of the PI3K/AKT/mTOR signaling pathway. Functionally, HOTAIR facilitates the expression of HSPA1A, a stress-inducible 70-kDa heat-shock protein [141,142], by sponging miR-449b-5p. In vitro and in vivo studies showed that miR-449b-5p overexpression or HSPA1A knockdown abrogated the ability of HOTAIR to increase BC growth under irradiation exposure and to support radiation resistance in BC [143]. Moreover, HOTAIR is a sponge of miR-129-5p and Frizzled7 (FZD7) [144] and contributes to BC progression by regulating the miR-129-5p/FZD7 axis, suggesting that HOTAIR may be a potential therapeutic target for BC [145]. Whether these interactions also play a role in CDR requires further investigations.

#### 2.6.3. LncRNA NEAT1 and Its Interaction with Different miRNAs

NEAT1 is involved in the organization of dynamic punctate compartments in the nucleus, so-called paraspeckles for gene transcription and splicing. Due to its oncogenic activity, it promotes EMT lung metastasis in BC cells. This is partly owed to its interaction with miRNAs. Thus, the mechanisms by which NEAT1 plays a critical role in cancers is through the NEAT1/miRNA/mRNA axes [146].

MiR-202, which is involved in the PTEN/AKT pathway, plays variable roles in tumorigenesis. It is assumed that its opposing features are mediated partly by its oncogene effectors. Thus, miR-202 may function as a tumor suppressor, the reduced levels of which in tumor tissues were found to be associated with the progression of different types of cancer. NEAT1 binds to miR-202 and inhibits its tumor-suppressive character, thus driving cancer progression [147].

Significant upregulation of lncRNA NEAT1 in BC tumors and cell lines was reported by Zhu et al. [103]. The silencing of lncRNA NEAT1 sensitized BC cells to paclitaxel. Bioinformatical analysis and luciferase assay were used to demonstrate that the tumor suppressor miR-23a-3p can be downregulated through sponging by NEAT1. The restoration of miR-23a-3p levels in NEAT1-overexpressing paclitaxel-resistant BC cells successfully overcame the NEAT1-promoted resistance. FOXA1, a regulator of steroid nuclear receptors to control transcription [49], was identified and validated as a direct target of miR-23a-3p. In addition, Li et al. found that upregulated NEAT1 promoted invasion through inducing EMT as well as resistance to 5-FU. They revealed a reciprocal repression between NEAT1 and miR-211, which targeted the downstream EMT-inducer high mobility group A2 (HMGA2) [99]. We identified NEAT1 and miR-204 as having a competitive impact on cell proliferation and apoptosis [131].

#### 2.6.4. LncRNA MALAT1 and Its Interaction with Different miRNAs

MALAT1 induces migration, invasion, angiogenesis, and metastasis, but it was described as a tumor suppressor, too [148]. It is also called NEAT2, and similar to NEAT1, it is mainly detected in nuclear speckles. In mice, loss of MALAT1 leads to the dysregulation of NEAT1. It has been reported to bind a number of miRNAs and regulate miRNA function as a ceRNA in various tumors [149]. In BC cells, MALAT1 binds to miR-1, miR-124, and miR-448, acting as a sponge to downregulate CDC42, a member of the Rho-GTPase family, as well as upregulating the cyclin-dependent kinase 4 CDK4 expression, leading to BC cell cycle progression, cell migration, and invasion [150,151,152].

In addition, Yue et al. [96] found that lncRNA MALAT1 was highly expressed in BC tissues and cells. Inhibition of the expression of MALAT1 was able to significantly suppress the proliferation, migration, and invasion of BC cells so sensitizing BC cells to doxorubicin. Bioinformatics and dual-luciferase reporter gene assay showed MALAT1 as a target of miR-570-3p that had low expression in BC tissues and cells. Thus, downregulation of MALAT1 led to upregulation of miR-570-3p. This resulted in not only the inhibition of the proliferation, metastasis, and invasion of BC cells but also augmented the sensitivity of BC cells to doxorubicin.

While the MALAT1–miRNA interactions described above have been supposed to promote cancer, MALAT1 has also been reported to exert tumor-suppressive effects by sponging miRNAs miR-17, 20a, and 106b, thereby decreasing the expression of epithelial cell adhesion molecule (EpCAM) in BC cells [153].

#### 2.6.5. LncRNA GAS5 and Its Interaction with miR-21

GAS5 is downregulated in various cancers and acts as a tumor suppressor in BC. It stimulates apoptosis in BC via diverse pathways and is also an important player in the regulation of signal pathways in BC, such as PI3K/AKT/mTOR, Wnt/β-catenin, and NF-κB signaling. Through epigenetic mechanisms, GAS5 can improve the sensitivity to multiple drugs and, thereby, the prognosis, suggesting it to be a promising target in the treatment of BC patients. In BC, GAS5 has been reported to bind to miR-21, miR-222, miR-221-3p, miR-196a-5p, and miR-378a-5p, indicating the presence of several sequence elements for miRNA binding in GAS5. The sequestering of these miRNAs results in the upregulation of a number of mRNAs of suppressor proteins, such as phosphatase and tensin homolog (PTEN), the human programmed cell death 4 (PDCD4), dickkopf-2 (Dkk2), forkhead-O1 (FoxO1), and suppressor of fused (SuFu). [63].

In addition, as detected by Li et al. [126] in trastuzumab-resistant SKBR-3 cells and tissue from trastuzumab-treated HER2+ BC patients, the expression of lncRNA GAS5 was decreased. The ability of GAS5 to suppress cancer proliferation is due to its acting as a molecular sponge for miR-21, leading to the de-repression of PTEN, the endogenous target of miR-21, whereas reduced GAS5 expression, which is associated with mTOR activation, suppressed PTEN.

#### 2.6.6. LncRNA ADAMTS9-AS2 and Its Interaction with miR-130a-5p

In BC tissues and tamoxifen-resistant BC cells, Shi et al. [118] detected a decreased expression of lncRNA ADAMTS9-AS2. Transient transfection led to overexpression of ADAMTS9-AS2. This, in turn, led to a reversal in the increased viability and decreased apoptosis that had been induced by miR-130a-5p mimics determined through the use of a dual-luciferase reporter gene assay. Confirmation of the binding between ADAMTS9-AS2 and miR-130a-5p was determined through downregulation of MALAT1. Hence, low expression of ADAMTS9-AS2 inhibited PTEN expression and increased tamoxifen resistance through targeting miR-130a-5p.

#### 2.6.7. LncRNA CYTOR and Its Interaction with miR-125a-5p

In their study, Liu et al. [117] identified cytoskeleton regulator CYTOR as the most significantly elevated lncRNA in tamoxifen-resistant MCF7 cells. In addition, they found a negative correlation between CYTOR and miR-125a-5p in BC tissues. Using bioinformatic analysis, they detected the binding of miR-125a-5p to CYTOR on regulating miR-125a-5p, CYTOR elevated the expression of serum response factor (SRF) [153], a transcription factor that regulates multiple genes involved in cell growth, migration, cytoskeletal organization, energy metabolism, and myogenesis [71]. It also activated the MAPK signaling pathway [64] to promote BC cell survival upon tamoxifen treatment. High expression of CYTOR found in BC patients’ tissues with no response to tamoxifen was positively correlated with SRF.

#### 2.6.8. LncRNA AGAP2-AS1 and Its Interaction with miR-15a-5p

Mesenchymal stem cells play a crucial role during the formation of drug resistance. The upregulation of AGAP2-AS1 was detected in resistant MSC-cultured cells by Han et al. [124], and knockdown of AGAP2-AS1 reversed the MSC-mediated trastuzumab resistance. However, AGAP2-AS1 regulated stemness and trastuzumab resistance via activating fatty acid oxidation (FAO). Mechanistically, AGAP2-AS1 was associated with human antigen R (HuR), and the AGAP2-AS1-HuR complex directly bound to carnitine palmitoyl transferase 1 (CPT1), raising its expression via improving RNA stability. As a ceRNA, AGAP2-AS1 sponged miR-15a-5p to release CPT1 mRNA. Thus, clinically, increased serum levels of AGAP2-AS1 may predict a poor response to trastuzumab treatment in BC patients.

#### 2.6.9. LncRNA AFAP1-AS1 and Its Interaction with miR-195 and miR-545

In their study, Wu et al. [98] found that formononetin restrained progression of TNBC by blocking the lncRNA AFAP1-AS1-miR-195/miR-545 axis. Using CCK8 and transwell assays, they showed (a) that silencing of lncRNA AFAP1-AS1 impaired chemoresistance, proliferation, migration, and invasion of TNBC MB-231 and BT cells, and (b) overexpression of miR-195 and miR-545, which were sponged and downregulated by AFAP1-AS1, reversed the stimulating effect of AFAP1-AS1 on proliferation, migration, invasion, and chemoresistance of TNBC cells. CDK4 and Raf-1 were downregulated by miR-195 and miR-545, respectively, but promoted by AFAP1-AS1. In addition, formononetin significantly decreased the levels of AFAP1-AS1, CDK4, and Raf-1 while increasing the levels of miR-195 and miR-545 in TNBC cells so alleviating TNBC malignancy.

#### 2.6.10. LncRNA MAFG-AS1 and Its Interaction with miR-339-5p

LncRNA MAFG-AS1 was shown to be upregulated in ER+ BC and tamoxifen-resistant MCF-7 cells by Feng et al. [121]. The estrogen-responsive MAFG-AS1 upregulated the cyclin-dependent kinase 2 (CDK2), an important cell cycle transition molecule [154], by sponging miR-339-5p, which stimulated ER+ BC proliferation. Tamoxifen resistance has been demonstrated to be the result of cross-talk between the ER signaling pathway and cell cycle, which appears to be controlled by MAFG-AS1 and CDK2.

#### 2.6.11. LncRNA HNF1A-AS1 and Its Interaction with miR-363

lncRNA HNF1A-AS1 silencing resulted in the reduction of proliferation and tamoxifen resistance of BC cells through the miR-363/SERTAD3 axis, together with the inactivation of the TGF-β/Smad pathway, as demonstrated by both in vitro and in vivo experiments [122]. MiR-363 was sponged by HNF1A-AS1, resulting in the promotion of the expression of the oncogene SERTAD3 [155]. In addition, either miR-363 downregulation or SERTAD3 upregulation led to a stimulation of tamoxifen resistance in BC cells.

#### 2.6.12. LncRNA LINC00461 and Its Interaction with miR-411-5p

In BC tissues, LINC00461 was upregulated relative to adjacent non-tumor tissues and its expression is associated with poor patient prognosis. In BC cells, knockdown of LINC00461 was able to inhibit cell proliferation. The levels of LINC00461 were higher in docetaxel-resistant than in non-resistant BC cell lines. Inversely, those of miR-411-5p were lower in docetaxel-resistant BC cell lines than in non-resistant cell lines. Accordingly, LINC00461 may serve as a ceRNA sponge for miR-411-5p. In both BC tissues and cell lines, miR-411-5p was downregulated, with levels negatively correlated with those of LINC00461 [156].

#### 2.6.13. LncRNA NONHSAT101069 and Its Interaction with miR-129-5p

In the BC specimens, BC cell lines, and epirubicin-resistant cell sublines, NONHSAT101069 was significantly overexpressed. Knockdown of NONHSAT101069 significantly repressed whereas overexpression of NONHSAT101069 promoted epirubicin resistance, migration, invasion, and the EMT process of BC cells both in vitro and in vivo. NONHSAT101069 acted as a ceRNA via sequestering miR-129-5p. Twist1 was a direct downstream protein of the NONHSAT101069/miR-129-5p axis in BC cells [97].

#### 2.6.14. LncRNA TINCR and Its Interaction with miR-125b

In BC, TINCR is associated with poor prognosis of patients who received trastuzumab therapy. Cell culture experiments showed that in contrast to untreated parental cells, TINCR was upregulated in trastuzumab-resistant cells. Knockdown of TINCR partially reversed resistance to trastuzumab and EMT by the regulation of miR-125b targeting both HER2 and Snail-1. Moreover, upregulation of TINCR was ascribed to transcriptional activation by H3K27 acetylation (H3K27ac) enrichment [127].

### 2.7. Therapies Using ncRNAs as Biomarkers and/or Targets and Their Challenges

A search in the Clinicaltrials.gov database presents numerous clinical trials involving ncRNAs in cancer. The majority of these trials use ncRNAs as biomarkers for cancer detection or as prognostic markers for disease outcome. Since ncRNAs circulate in different body fluids, e.g., blood and urine, or in exosomes, the investigation is less invasive than that of the tumor. During therapy the deregulation of ncRNAs in the liquid biopsies may predict survival, metastasis, or therapy response [157]. Hence, the determination of ncRNA levels may serve as potential diagnostic and prognostic clinical parameters and account for the numerous clinical studies.

NcRNAs may be used as therapeutic targets or agent for BC. The development of ncRNA-based therapies aims at normalizing the aberrant level of ncRNAs in cancer. Pertubations of the RNA expression contribute to tumor progression and development of CDR resulting in the downregulation of tumor-suppressive ncRNAs and the upregulation of oncogenic ncRNAs. Restoring the normal levels of ncRNAs by promoting miRNA tumor-suppressive function and inhibiting the oncogenic function may be used as a cancer treatment strategy to ensure a normal regulatory network of signals. NcRNA-based precision medicine relies on the use of either synthetic miRNA/lncRNA-like molecules. These include mimics and cloned expression vectors that restore either miRNA expression or anti-ncRNAs that are known as antagomirs including antisense oligonucleotides, miRNA sponges, and anti-ncRNA peptides that either inhibit miRNA expression or degrade mRNAs [158,159].

However, clinical studies on ncRNAs as therapeutic targets are less promising. In this respect, a multicenter phase 1 study on miR-RX34 (see ClinicalTrials.gov: NCT01829971, NCT02862145) sponsored by Mirna Therapeutics investigated miR-34 in patients with unresectable primary liver cancer, advanced or metastatic cancer with or without liver involvement, or hematologic malignancies. This tumor suppressive miRNA is a direct target of p53 and regulates the expression of several oncogenes. Whereas the previous study showed good results, the phase 1b study unfortunately showed immune-related toxicities in many patients, and thus the trial had to be terminated [160].

The adverse effects in the patients caused by the delivery of ncRNAs can be explained by their multiple function. A single ncRNA has binding activity to numerous targets and so has the potential to modulate the expression of multiple genes that in turn influence numerous signaling pathways [161]. Moreover, due to the crosstalk between the various signaling pathways, ncRNAs can theoretically affect further multiple interconnected signaling pathways at once. In addition, ncRNAs seem to possess dual functions, acting either as both a tumor suppressor and as an oncogene [162]. miR-200 is an example in BC that has both tumor suppressive and oncogenic behavior. It inhibits EMT and therefore prevents initiation of BC metastasis. However, upregulated levels of this miRNA were detected in late-stage BC and may promote distant metastasis [163]. Besides this multitasking character, the mutual interactions of ncRNAs, e.g., lncRNAs with miRNAs, that modulate the function of both ncRNAs, have to be considered. These findings demonstrate the need for continuous research to better understand the character and role of ncRNAs in the regulatory network cancer.

The delivery method for RNA-based drugs has to be also taken into account and optimized, and this can often be a challenging process [164]. Due to their character to mediate cell-to-cell communication and cargo transport, polymer-based vesicles, e.g., exosomes, can serve as delivery systems of ncRNAs. As described in detail by Wang et al. [165], there are some challenges to overcome for their clinical application, e.g., cell-uptake, drug loading, drug release, and in vivo distribution. Thus, the development of successful targeted therapies has some challenges. They have also to overcome the rapid degradation of ncRNAs or anti-ncRNAs by cellular nucleases, poor cellular uptake, non-specific binding to bystander mRNAs eliciting unspecific effects plus toxicity, and/or unfavorable immune response.

There are numerous candidate miRNAs to overcome cancer detection in BC that can be used in precision medicine. For example, the upregulation of miR-20a and downregulation of miR-451 after the second cycle of neoadjuvant chemotherapy predicted resistance to treatment of hormone receptor-positive/HER2-negative BC patients [166]. In TNBC patients who received neoadjuvant cisplatin/doxorubicin-based chemotherapy, miR-145-5p was downregulated in patients who achieved a complete pathological response [167]. In another study analyzing luminal BC patients who received neoadjuvant chemotherapy, the level of miR-145 was significantly lower in responders compared to non-responders [168].

Although the functions of lncRNAs have been well analyzed in cancer, they are rarely translated into clinical practice in contrast to miRNAs. However, in a comprehensive profiling study with training and testing datasets, a three-lncRNA signature (AK291479, U79293, and BC032585) was identified for a cohort of 1102 BC patients to predict a pathological complete response after neoadjuvant chemotherapy [169]. LncRNAs can also be used to cluster BC patients into subgroups, and five transcripts of the MALAT1 gene were specifically upregulated in resistant patients [76].

To date, various RNA-based therapies (NCT01722851, NCT0295020, NCT0159828, NCT0477187) have been developed, with some being approved by the U.S. Food and Drug Administration (FDA).

## 3. Conclusions

An understanding of the aberrant signal transduction pathways that form the basis for tumor progression and metastasis have resulted in the use of targeted therapies that lead to an inhibition of oncogenic signals. However, such aberrant signal transduction pathways may also lead to CDR. In clinical settings, CDR is a prime reason resulting in the failure of BC therapy. CDR is a complex process in which many factors are involved. These include the interplay between lncRNAs and miRNAs. One signaling pathway can be disturbed by several lncRNAs or miRNAs, whilst a single lncRNA or miRNA can be involved in affecting several pathways. lncRNAs can act as ceRNAs. In this way, they are able to affect the activity and function of miRNA by post-transcriptional regulation. Thus, lncRNAs can de-repress gene expression when they compete with miRNAs in their interaction with shared target mRNAs. In this regard, the ceRNA theory was introduced by Salmena et al. [23] who reported that miRNAs can build a bridge for the communication among various types of RNAs, such as lncRNAs. On the one hand, lncRNAs may be the precursor of miRNAs, and hence produce miRNAs, leading to repression of their target mRNAs. This miRNA/lncRNA cross-talk modulates the gene expression driving pathological processes, such as cancer and drug resistance. More recently, and using the approach of Salmena et al. [23], Yang et al. [170] identified the small-molecule drugs together with their affected lncRNAs. Liu et al. [171] extended this approach by devising 15 ceRNA networks involving 15 anti-cancer drug categories and implicating 217 lncRNAs, 158 miRNAs, and 1389 protein-coding genes. They further established the presence of “an intersection ceRNA network (ICN) across the 15 anti-cancer drug categories”. In addition, these researchers found that some of the genes present in the ICN were either up- or downregulated in tumors.

To date, the exact mechanism of this network regulating drug resistance is still not fully defined. Therefore, before developing a targeted therapy involving either miRNAs or lncRNAs, their multiple behaviors and interplay in both time and cellular location should be analyzed. This also means that tumor suppressive features of some miRNAs cannot affect the tumor progression because they may be sponged by lncRNAs, and therefore, the entire context of both molecules should be considered.

In summary, the development of a number of targeted agents with minimal adverse effects has been improved, but efficient options for BC treatment and overcoming drug resistance still remain limited. Thus, elucidating and overcoming the pathogenesis of drug resistance and the role of ncRNAs involved in the regulatory network remains an important task for the future.

## Figures and Tables

**Figure 1 ijms-24-08095-f001:**
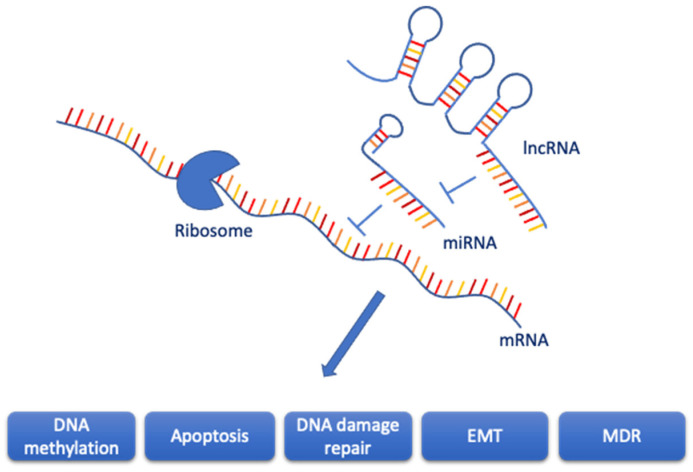
Interplay of lncRNA and miRNA in the modulation of RNA expression in different biological processes.

**Figure 2 ijms-24-08095-f002:**
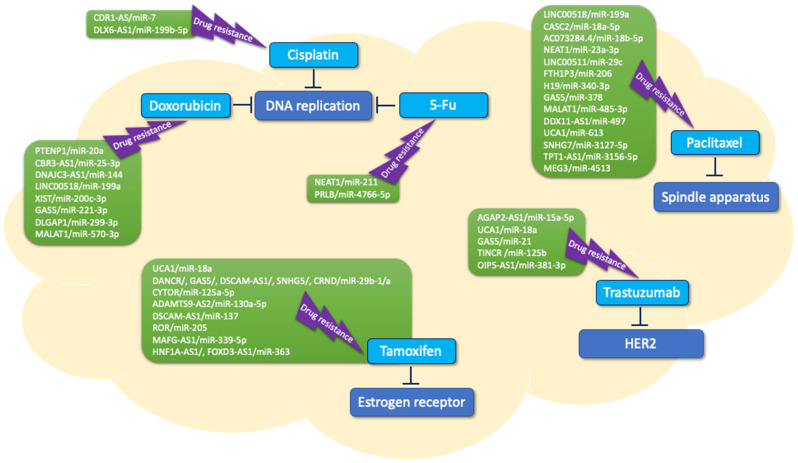
Interplay of lncRNA and miRNA in resistance of different drugs in breast cancer.

**Table 1 ijms-24-08095-t001:** Chemotherapeutic drugs used in the treatment of breast cancer patients.

Drug	Group	Character	Function
Cisplatin		Cytostatic	Inhibition of DNA replication
Docetaxel	Taxanes	Cytostatic	Derivative of paclitaxel
			Binding to microtubules suppressing the function of the spindle apparatus
Doxorubicin	Anthracyclines	Intercalant	Topoisomerase II inhibitor
Adriamycin			Blocking DNA/RNA synthesis
Epirubicin	Anthracyclines	Intercalant	Stereoisomer of doxurubicin
			Topoisomerase II inhibitor
Formononetin	Isoflavones	Phytoestrogen	Estrogen activity
5-FU	Antimetabolite	Pyrimidine analog	Inhibition of replication
Paclitaxel	Taxanes	Cytostatic	Binding to β-tubulin inhibiting the degradation of spindle fibers, blocking mitotic cell division in the G2, M phase
Rapamycin	Immunosuppressor		Inhibition of mTOR
Tamoxifen	ER antagonist	ER modulator	Inhibition of ER and stimulation of PR
Trastuzumab	Antibody	HER2 modulator	Inhibition of HER2

ER, estrogen receptor; 5-FU, 5-fluorouracil; HER2, human epidermal growth factor receptor; PR, progesterone receptor; mTOR, mammalian target of rapamycin.

## Data Availability

Not applicable.

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
