# Peer review of "Interplay between LncRNAs and microRNAs in Breast Cancer"

_ijms, 2023, doi:10.3390/ijms24098095_

Round 1

Reviewer 1 Report

This review is interesting, but it needs extensive revision. The current version is not very clear in delivering the author's message.

1. I suggest that section 2 should not be titled 'Results' because this is a review and the author did not perform any experiments.

 2.  The involvement of lncRNAs and miRNAs in signaling pathways and drug resistance should be combined with sections 2.1 and 2.2. Otherwise, it may confuse readers because sections 2.1 and 2.2 do not explicitly discuss the link between lncRNAs and miRNAs.

3.  Section 2.5 is very long and would benefit from subheadings to help readers navigate through it.

4.  It would be helpful to discuss whether lncRNAs and miRNAs regulate angiogenesis.

5. I suggest creating a new section titled 'Mechanism of lncRNA-miRNA Interactions.

6.  The author should add a section discussing non-coding RNA delivery systems.

7.  The article should include a discussion of current challenges and limitations for non-coding RNA as therapeutic targets.

Author Response

Manuscript ID: ijms-2295722

Point by point reply to the comments raised by the reviewers with respect to the manuscript “Interplay between lncRNAs and microRNAs in breast cancer detection rates” by Heidi Schwarzenbach and Peter B. Gahan

The reviewers’ comments are numbered and in regular font, while our reply is in italics. Modifications made in the revised manuscript are highlighted in red.

We would like to thank the reviewers for their time and effort to provide their reviews. We have carefully and substantially revised the manuscript according to their suggestions.

Reviewer 1

  1. I suggest that section 2 should not be titled 'Results' because this is a review and the author

did not perform any experiments.

We fully agree with the Reviewer. However, we have followed the journal guidelines that  require this title. We will remove it and discusss further with the Editor if necessary.

  1. The involvement of lncRNAs and miRNAs in signaling pathways and drug resistance should

be combined with sections 2.1 and 2.2. Otherwise, it may confuse readers because sections 2.1

and 2.2 do not explicitly discuss the link between lncRNAs and miRNAs.

We have combined sections 2.1 and 2.2 (page 3, line 90, 2.1. i.e Effective drugs and important signaling pathways in breast cancer, with subheadings 2.1.1.-2.1.10., pages 4-7).

  1. Section 2.5 is very long and would benefit from subheadings to help readers navigate through it.

Section “2.5. CeRNA network in breast cancer” now contains subheadings (pages 11-17, 2.5.1.-2.5.13.).

  1. It would be helpful to discuss whether lncRNAs and miRNAs regulate angiogenesis.

A new section “2.4. Involvement of lncRNAs and miRNAs in angiogenesis” was added to the manuscript (pages 10-11).

  1. I suggest creating a new section titled 'Mechanism of lncRNA-miRNA Interactions.

We have added a new section “2.2.5. Mechanism of lncRNA-miRNA interactions” to the manuscript (page 8).

  1. The author should add a section discussing non-coding RNA delivery systems.

We have added a new section “2.3. NcRNA delivery systems” to the manuscript (pagers 9-10).

  1. The article should include a discussion of current challenges and limitations for non-coding

RNA as therapeutic targets.

We have included a discussion on the relevance, challenges and limitations of ncRNAs (see also 2.6. Therapies using ncRNAs as biomarkers and/or targets and their challenges, pages 17-18).

Reviewer 2 Report

In the manuscript titled “Interplay between lncRNAs and microRNAs in breast cancer detection rates”, the authors review the crosstalk between lncRNAs and miRNAs in breast cancer pathology and drug resistance. They briefly mention about drugs used against breast cancer with their resistance pathways and thoroughly discuss the involvement of lncRNA-miRNAs axis in these resistance pathways. It would be better if the authors emphasized the significance and future perspectives of the topic in depth in the conclusion section.

Major comments:

  • Despite its title, the article does not really discuss about “cancer detection rates”. Instead, it provides a detailed summary for interplay between lncRNAs and miRNAs in resistance mechanisms in breast cancer.
  • Page 14. Paragraphs starting from 575 review a different aspect from the previous ones. These can be explained under another sub-header rather than “5. Interplay between lncRNAs and miRNAs” section.

Minor comments:

  • Abstract first sentence can be made more clear by combining Line 9-10; such as “Although lncRNAs are known to be precursors of miRNAs, they frequently act as competing endogoneous RNAs (ceRNAs), yet, still their interplay with miRNA is not well-known.”
  • Lines 23-24-25. Three groups can be written more explicitly by writing 1) ER +/- 2) and 3)
  • Line 37. “…. BC – …” sentence has – sign which is not understood.
  • Lines 74-75. Figure 1 caption needs more explanations especially about abbreviations.
  • Line 142. “ Immediate hypersensitivity…” sentence is not clear. It should be explained more.
  • Line 203. In the sentence starting with “As detailed described in a review by …”, use of both verbs “detailed” and “described” is redundant. Can be changed to either “As described in detailed”, “As described” or “As detailed”.
  • Line 214. It is not clear what author meant by the term “detection treatment failure”.
  • Line 224. The word “…de-repression…” is interesting and reminds depression so it might cause some confusion. Instead, repressed again etc. can be used.
  • Line 243. Typo “TGB-B” should be “TGF-β”.
  • Line 251: LINC00665 does not have any explanation. Is it lncRNA or miRNA?
  • Line 294. There can be a subtitle such as “How are miRNAs made?”
  • Lines 316-317. Typo “lnRNAs” should be “lncRNAs”.
  • Line 322. There seems to be a double space after the full stop.
  • Line 329. The authors say “above” but Table 2 is located below the text.
  • Line 333. Typo “outcome play a role in cancer” should be “outcome plays”.
  • Lines 371-372. “Wnt/β-catenin” skipped to next line but should be following “Dickkopf 2” in line 371.
  • Lines 395-397. Not in line. There is extra tab.
  • Lines 400-403. There is not enough space between “tamoxifen” and “molecule/pathway” columns.
  • Line 630. In the sentence “… these workers found …”, “workers” can be replaced with “authors”, “researchers” or “investigators”.

Author Response

Manuscript ID: ijms-2295722

Point by point reply to the comments raised by the reviewers with respect to the manuscript “Interplay between lncRNAs and microRNAs in breast cancer detection rates” by Heidi Schwarzenbach and Peter B. Gahan

The reviewers’ comments are numbered and in regular font, while our reply is in italics. Modifications made in the revised manuscript are highlighted in red.

We would like to thank the reviewers for their time and effort to provide their reviews. We have carefully and substantially revised the manuscript according to their suggestions.

Reviewer 2

  1. Despite its title, the article does not really discuss about “cancer detection rates”. Instead, it provides a detailed summary for interplay between lncRNAs and miRNAs in resistance mechanisms in breast cancer.

We have removed “cancer detection rates” from  the title.

  1. Page 14. Paragraphs starting from 575 review a different aspect from the previous ones. These can be explained under another sub-header rather than “5. Interplay between lncRNAs and miRNAs” section.

We changed the title “Interplay between lncRNAs and miRNAs” and replaced it by “CeRNA network in breast cancer”. This section 2.5. was also amended by subheadings (2.5.1.-2.5.13., pages 11-17).

  1. Abstract first sentence can be made more clear by combining Line 9-10; such as “Although lncRNAs are known to be precursors of miRNAs, they frequently act as competing endogoneous RNAs (ceRNAs), yet, still their interplay with miRNA is not well-known.”

We adopted the Reviewer’s sentence (page 1, Abstract).

4.Lines 23-24-25. Three groups can be written more explicitly by writing 1) ER +/- 2) and 3)

We have explicitely described the three groups (page 1).

  1. Line 37. “…. BC – …” sentence has – sign which is not understood.

We canceled the sign “-“ (page 1, line 37).

  1. Lines 74-75. Figure 1 caption needs more explanations especially about abbreviations.

Figure 1 is now explained in more detail (page 2).

  1. Line 142. “ Immediate hypersensitivity…” sentence is not clear. It should be explained more.

The sentence now includes a clearer explanation (page 5).

  1. Line 203. In the sentence starting with “As detailed described in a review by …”, use of both

verbs “detailed” and “described” is redundant. Can be changed to either “As described in

detailed”, “As described” or “As detailed”.

We canceled “detailed” and the sentence now starts with “As described in a review …” (page 5, line 220).

  1. Line 214. It is not clear what author meant by the term “detection treatment failure”.

We canceled “detection” (page 6, line 230-231).

  1. Line 224. The word “…de-repression…” is interesting and reminds depression so it might cause some confusion. Instead, repressed again etc. can be used.

We replaced “de-repression” by “expression” (page 6, line 241).

  1. Line 243. Typo “TGB-B” should be “TGF-β”.

We replaced “TGB-B” by “TGF-β” (page 6, line 260).

  1. Line 251: LINC00665 does not have any explanation. Is it lncRNA or miRNA?

We have added lncRNA before LINC00665 (page 6, line 269).

  1. Line 294. There can be a subtitle such as “How are miRNAs made?”

As suggested, we subtitled the “2.2. LncRNAs and miRNAs” (2.2.1.-2.2.4., pages 7-8).

  1. Lines 316-317. Typo “lnRNAs” should be “lncRNAs”.

We corrected “lnRNAs” to “lncRNAs” (page 8, line 348).

  1. Line 322. There seems to be a double space after the full stop.

We removed the double space before “This approach …” (page 8, line 376).

  1. Line 329. The authors say “above” but Table 2 is located below the text.

“above” related to the signaling pathways which are described above. So as not to confuse the readership, we corrected it to “and as shown in Table 2 below” (page 10, line 439).

  1. Line 333. Typo “outcome play a role in cancer” should be “outcome plays” (page, line).

We changed “play” to “plays” (page 10, line 443).

  1. Lines 371-372. “Wnt/β-catenin” skipped to next line but should be following “Dickkopf 2” in line 371.

We corrected the whole Table 2 (pages 11-13).

  1. Lines 395-397. Not in line. There is extra tab.

We corrected the whole Table 2 (pages 11-13).

  1. Lines 400-403. There is not enough space between “tamoxifen” and “molecule/pathway” columns.

We corrected the formatting of the whole Table 2 (pages 11-13).

  1. Line 630. In the sentence “… these workers found …”, “workers” can be replaced with “authors”, “researchers” or “investigators”.

“workers” was replaced by “researchers” (page 19, line 834).

Round 2

Reviewer 1 Report

The current version has shown some improvement compared to the previous one, but it still needs further revisions before it can be published.

1.      It is unclear what is meant by the phrase "Results/MEK/ERK, Wnt/β-catenin, and transforming growth factor beta (TGF-β)/Smad 90 signaling".

2.      The signaling pathways described by the author are too lengthy and not directly related to the study's title, "Interplay between lncRNAs and microRNAs in breast cancer". The author should focus on the signaling pathways reported to have interplay between lncRNAs and microRNAs. The current version seems to include any signaling pathway reported in lncRNAs and microRNAs, which does not necessarily mean that they have interplay.

3.      It may not be necessary to provide an extensive paragraph on the production and functions of lncRNA and microRNA. If the author wishes to explain the interaction mechanism between these two, it should be discussed before the signaling pathways.

4.      Section 2.5 is confusing as the title mentions lncRNAs such as H19, HOTAIR, and NEAT1, but the content is about their interplay with microRNA. Moreover, the author has provided numerous examples of lncRNA and microRNA interplay without explaining them in-depth. It would be better to choose the top five and provide a more detailed explanation of how they interact.

Author Response

Manuscript ID: ijms-2295722

Point by point reply to the comments raised by the reviewers with respect to the manuscript “Interplay between lncRNAs and microRNAs in breast cancer detection rates” by Heidi Schwarzenbach and Peter B. Gahan

The reviewers’ comments are numbered and in regular font, while our reply is in italics. Modifications made in the revised manuscript are highlighted in red.

We would like to thank the reviewer for their time and effort to provide their reviews. We have carefully and substantially revised the manuscript according to their suggestions.

Reviewer 1

  1. It is unclear what is meant by the phrase "Results/MEK/ERK, Wnt/β-catenin, and transforming growth factor beta (TGF-β)/Smad 90 signaling".

We fully agree with the reviewer. It was an oversight and is very annoying (page 3).

  1. The signaling pathways described by the author are too lengthy and not directly related to the study's title, "Interplay between lncRNAs and microRNAs in breast cancer". The author should focus on the signaling pathways reported to have interplay between lncRNAs and microRNAs. The current version seems to include any signaling pathway reported in lncRNAs and microRNAs, which does not necessarily mean that they have interplay.

We have shortened the paragraph about the signaling pathways and removed the paragraph “2.1.10. Notch”. The other signaling pathways relate to the focus of the manuscript (page 9).

  1. It may not be necessary to provide an extensive paragraph on the production and functions of lncRNA and microRNA. If the author wishes to explain the interaction mechanism between these two, it should be discussed before the signaling pathways.

We have provided a paragraph on the production and functions of lncRNAs and miRNAs because Reviewer 2 wanted such an extensive paragraph. We have moved this paragraph before the signaling pathways (page 3-4).

  1. Section 2.5 is confusing as the title mentions lncRNAs such as H19, HOTAIR, and NEAT1, but the content is about their interplay with microRNA. Moreover, the author has provided numerous examples of lncRNA and microRNA interplay without explaining them in-depth. It would be better to choose the top five and provide a more detailed explanation of how they interact.

To avoid such confusion, we added to the subsections of section 2.5. (now 2.6.)“and its interaction with … miRNAs”. As suggested by the reviewer, we chose five lncRNAs, H19, HOTAIR, NEAT1, MALAT1 and GAS5 since they interact with several miRNAs in breast cancer and because about the interplay of the OTHER lncRNAs is less known in breast cancer (page 14-18). To meet the comments of each reviewer, we did not remove the subsections of the OTHER lncRNAs. If this reviewer demands it, then we will have to ask for permission from the reviewer 2.

Round 3

Reviewer 1 Report

Author has addressed my concern 

Author Response

We corrected the spelling and removed both references which were double cited. We also added both references which are wished by the editor to the manuscript (marked in red).